# Formulation and Characterization of New Polymeric Systems Based on Chitosan and Xanthine Derivatives with Thiazolidin-4-One Scaffold

**DOI:** 10.3390/ma12040558

**Published:** 2019-02-13

**Authors:** Sandra Madalina Constantin, Frederic Buron, Sylvain Routier, Ioana Mirela Vasincu, Maria Apotrosoaei, Florentina Lupașcu, Luminița Confederat, Cristina Tuchilus, Marta Teodora Constantin, Alexandru Sava, Lenuţa Profire

**Affiliations:** 1Department of Pharmaceutical Chemistry, Faculty of Pharmacy, University of Medicine and Pharmacy “Grigore T. Popa”, 16 University Street, 700115 Iasi, Romania; constantin.sandra@umfiasi.ro (S.M.C.); ioanageangalau@yahoo.com (I.M.V.); apotrosoaei.maria@umfiasi.ro (M.A.); florentina-geanina.l@umfiasi.ro (F.L.); 2Institut de Chimie Organique et Analytique, Univ Orleans, CNRS, ICOA, UMR 7311, F-45067 Orléans, France; frederic.buron@univ-orleans.fr (F.B.); sylvain.routier@univ-orleans.fr (S.R.); 3Department of Microbiology, Faculty of Pharmacy, University of Medicine and Pharmacy “Grigore T. Popa”, 16 University Street, 700115 Iasi, Romania; luminita.confederat@umfiasi.ro (L.C.); cristina.tuchilus@umfiasi.ro (C.T.); 4University of Kent, School of Physical Sciences, Canterbury CT2 7NH, UK; marta.constantin97@gmail.com; 5Department of Analytical Chemistry; Faculty of Pharmacy, University of Medicine and Pharmacy “Grigore T. Popa”, 16 University Street, 700115 Iasi, Romania; alexandru.i.sava@umfiasi.ro

**Keywords:** thiazolidine-4-one scaffold, chitosan, polymeric systems, antibacterial activity

## Abstract

In the past many research studies have focused on the thiazolidine-4-one scaffold, due to the important biological effects associated with its heterocycle. This scaffold is present in the structure of many synthetic compounds, which showed significant biological effects such as antimicrobial, antifungal, antioxidant, anti-inflammatory, analgesic, antidiabetic effects. It was also identified in natural compounds, such as actithiazic acid, isolated from *Streptomyces* strains. Starting from this scaffold new xanthine derivatives have been synthetized and evaluated for their antibacterial and antifungal effects. The antibacterial action was investigated against Gram positive (*Staphyloccoccus aureus* ATCC 25923, *Sarcina lutea* ATCC 9341) and Gram negative (*Escherichia coli* ATCC 25922) bacterial strains. The antifungal potential was investigated against Candida spp. (*Candida albicans* ATCC 10231, *Candida glabrata* ATCC MYA 2950, *Candida parapsilosis* ATCC 22019). In order to improve the antimicrobial activity, the most active xanthine derivatives with thiazolidine-4-one scaffold (XTDs: 6c, 6e, 6f, 6k) were included in a chitosan based polymeric matrix (CS). The developed polymeric systems (CS-XTDs) were characterized in terms of morphological (aspect, particle size), physic-chemical properties (swelling degree), antibacterial and antifungal activities, toxicity, and biological functions (bioactive compounds loading, entrapment efficiency). The presence of xanthine-thiazolidine-4-one derivatives into the chitosan matrix was confirmed using Fourier transform infrared (FT-IR) analysis. The size of developed polymeric systems, CS-XTDs, ranged between 614 µm and 855 µm, in a dry state. The XTDs were encapsulated into the chitosan matrix with very good loading efficiency, the highest entrapment efficiency being recorded for CS-6k, which ranged between 87.86 ± 1.25% and 93.91 ± 1.41%, depending of the concentration of 6k. The CS-XTDs systems showed an improved antimicrobial effect with respect to the corresponding XTDs. Good results were obtained for CS-6f, for which the effects on *Staphylococcus aureus* ATCC 25923 (21.2 ± 0.43 mm) and *Sarcina lutea* ATCC 9341 (25.1 ± 0.28 mm) were comparable with those of ciprofloxacin (25.1 ± 0.08 mm/25.0 ± 0.1 mm), which were used as the control. The CS-6f showed a notable antifungal effect, especially on *Candida parapsilosis* ATCC 22019 (18.4 ± 0.42 mm), the effect being comparable to those of nystatin (20.1 ± 0.09 mm), used as the control. Based on the obtained results these polymeric systems, consisting of thiazolidine-4-one derivatives loaded with chitosan microparticles, could have important applications in the food field as multifunctional (antimicrobial, antifungal, antioxidant) packaging materials.

## 1. Introduction

In the last time the xanthine scaffold was widely investigated based on its high biological potential. Some of the most important derivatives are caffeine, theophyline, theobromine, and paraxanthine [1]. These compounds are known for important biological effects such as bronchodilator, diuretic, anti-inflammatory and lipolytic [2,3]. New biological active molecules with a xanthine structure have been developed and evaluated for potential applications in asthma [4], diabetes mellitus [5], cancer [6], microbial infections [7] and neurodegenerative diseases such as Parkinson [8] and Alzheimer [2]. 

In the medicinal chemistry the thiazolidine heterocycle is also known as an important scaffold that is used to modulate the classical structures in order to improve their biological effects and to induce other new ones. For the first time this structure was identified as an actithiazic acid, a natural compound isolated from *Streptomyces spp.*, which showed high and specific action against *Mycobacterium tuberculosis* [9]. Later, many thiazolidine derivatives were synthesized and evaluated for their antimicrobial effect [10,11,12]. It was shown that the antimicrobial effects are in close correlation with the nature and position of substituents on the heterocycle. For example, the presence of arylazo, phenylhydrazono or sulfamoylphenylazo on the heterocycle was associated with improved antimicrobial effects [13].

In a previous study the researchers focused their studies on micro and nanotechnology as new strategies for increasing solubility, improving the biological properties, and decreasing the toxicity of the drugs. Some of the most important applications are in the regenerative medicine or tissue engineering area, as biosensors or as controlled drug release systems. The polymeric systems such as nano and microparticels are largely used as drug delivery systems due to their small particle size and large surface area [14]. Lately, chitosan, a natural polymer derived from chitin, has attracted the interest of many researchers due to its important and specific properties, such as biocompatibility, biodegradability, and reduced toxicity [15]. This biopolymer is considered the largest biomaterial after cellulose, having important applications in both the pharmaceutical and the food industry [16,17]. It is a linear policationic polysaccharide, copolymer of D-glucosamine and N-acetyl-D-glucosamine linked in 1-4 positions [16]. The amino groups on chitosan allow it to react with varied types of reagents to introduce different functional groups to chitosan. This polymer has important biological effects such as antimicrobial [18,19], antidiabetic [20,21,22], antitumor [23], antioxidant [24,25], anti-inflammatory [24], antiulceros [26], and not least, hypocholesterolemic [27]. The antimicrobial effect is the result of the interaction between the negatively charged bacterial cell membranes and the positively charged amino groups of chitosan, being directly proportional to the molecular weight and inversely proportional to the pH value. The studies showed also that *Gram-positive* bacterial strains are more susceptible to chitosan than *Gram-negative* ones [28].

Referring to the food industry, it is known that the packaging components have an important role for preservation of food as well as for environment protection, in a previous study the researchers focused on the use of renewable material and on the development of active packaging in order to increase the quality and safety of food [29]. The embedded food technology based on antimicrobial agents represents nowadays a challenge for researchers being strengthened also by the few marked products [30]. To obtain active packaging the antimicrobial agents could be included in the polymer matrix or the polymer surface could be covered with different antimicrobial agents [31,32]. Knowing that the microbial growth occurs mainly on the food surface, this new strategy involves slow diffusion of the antimicrobial agent from the package material, such as polymer films [33]. The controlled release packaging is a new approach for the food industry. The literature reported various packaging systems based on chitosan or embedded chitosan with different antimicrobial agents [34,35,36,37]. The studies highlight the important role chitosan plays as a plasticizer and it is frequently used in different composite systems with potential application in the food industry in order to increase the mechanical strength of the developed structure [37]. 

In this study we report the preparation, physic-chemical characterization, and biological evaluation of new polymeric systems based on chitosan, which have been loaded with new xanthine derivatives with a thiazolidine-4-one scaffold. Based on the unique properties of chitosan (biodegradability, biocompatibility) and of the biological effects of xanthine derivatives with a thiazolidine-4-one scaffold, the developed polymeric systems could have important applications in the food industry as active packaging materials and also in the medical field. 

## 2. Materials and Methods 

### 2.1. Materials

Chitosan (CS, molecular weight 190–310 kDa, 75–85% deacetylation degree), pentasodium tripolyphosphate (TPP), disodium hydrogen phosphate, sodium dihydrogen phosphate, hydrochloric acid 37%, acetic acid (min. 99,8%, p.a. ACS reagent), dimethyl sulfoxide (DMSO, p.a., ACS regent), Mueller–Hinton agar medium (Oxoid), Sabouraud agar medium were purchased from Sigma Aldrich. Xanthine derivatives with thiazolidine-4-one scaffold (XTDs: 6c, 6e, 6f, 6k) have been synthesized by our research group and were reported in a previous paper [38]. Bacterial strains (*Staphyloccoccus aureus* ATCC 25923, *Sarcina lutea* ATCC 9341, *Escherichia coli* ATCC 25922) and yeast strains (*Candida albicans* ATCC 10231, *Candida glabrata* ATCC MYA 2950, *Candida parapsilosis* ATCC 22019) were obtained from the Department of Microbiology, “Grigore T. Popa” University of Medicine and Pharmacy, Iasi, Romania. Ampicillin (30 µg/disc), ciprofloxacin (25 µg/disc) and nystatin (100 µg/disc) were used as positive controls for antibacterial and antifungal activity respectively. Swiss mice were purchased from the Biobase of “Grigore T. Popa” University of Medicine and Pharmacy, Iasi, Romania.

### 2.2. Development of New Polymeric Systems Based on Chitosan

#### 2.2.1. Preparation of Chitosan Microparticles

Chitosan microparticles were prepared by the ionic gelation method. Briefly, chitosan (CS, 1.2–1.7 g) was dispersed in 100 mL of acetic acid 1% (*v*/*v*) and the mixture was left under stirring overnight. 3 mL of chitosan solution were dropped, using a syringe needle (26 G; 0.45 mm × 16 mm), into 20 mL of TPP (2%–3%), under light stirring and the mixture was left at room temperature, under stirring for 4–8 h in order to achieve an efficient reticulation. After 24 h the formed chitosan beads were separated from the TPP solution, washed three times with distillated water, and air-dried at room temperature.

#### 2.2.2. Preparation of Chitosan Microparticles Loaded with Xanthine Derivatives with Thiazolidine-4-One Scaffold (CS-XTDs)

The xanthine derivatives with thiazolidine-4-one scaffold (XTDs: 6c, 6e, 6f, 6k) were loaded into chitosan microparticles by the ionic gelation method [5]. Briefly, different amounts of XTD (9 mg, 12 mg, 15 mg) were dissolved in 0.3 mL of DMSO and mixed with 3 mL of chitosan solution (1.2%–1.7%, *v*/*v*) in order to obtain three different concentrations (3 mg/mL; 4 mg/mL, 5 mg/mL) for each derivative (6c, 6e, 6f, 6k). The resulted mixtures were dropped, using a syringe needle (26 G; 0.45 mm × 16 mm), into 20 mL of TPP (2%–3%), under light stirring and the mixture was left at room temperature, under stirring for 4–8 h to achieve an efficient reticulation. After 24 h the formed chitosan loaded beads (CS-XTDs) were separated fromTPP solution, washed with distillated water three times, and air-dried at room temperature.

### 2.3. Characterization of Chitosan Microparticles Loaded with Xanthine Derivatives with Thiazolidine-4-one Scaffold (CS-XTDs)

#### 2.3.1. Particle Size and Morphology

The CS-XTDs microparticle size (in wet and dry state) was measured using a Leica DM750 microscope (Wetzlar, Germany) equipped with a video model ICC50 W0366. The values were recorded with 10× objective by LAS EZ program. The morphology of the microparticles was studied using a scanning electronic microscope (SEM), Vega II SBH model, produced by the Tescan Company (Brno, Czech Republic). All measurements were performed in triplicate and the results are expressed as mean size ± standard deviation.

#### 2.3.2. Swelling Degree (SD)

The swelling studies were carried out using distilled water and simulated gastric fluid (SGF, pH 2.2) at 37 °C (in a thermostated water bath) according to the literature references [5,39]. Briefly, an exact amount of CS-XTDs microparticles was immersed into the media. At different times microparticles were removed from media (water and SGF respectively), dried quickly with filter paper, and weighted (W_1_). At the end of the study the microparticles were dried again and weighted (W_2_). All tests were carried out in triplicate and the results are expressed as mean SD ± standard deviation. The analysis of Variance (ANOVA) was used to analyze the experimental data. Statistical significance was set to *p* value ≤ 0.05.

The swelling degree (SD) at different times was calculated using the following formula:SD (%) = W_1_ − W_2_/W_2_ × 100(1)
where: W_1_—the weight of the swollen CS-XTDs microparticles; W_2_—the weight of the dried CS-XTDs microparticles.

#### 2.3.3. Drug Loading and Entrapment Efficiency

The loading efficiency of XTDs (6c, 6e, 6f, 6k) into the chitosan matrix was analyzed using a GBC Cintra 2010 UV-VIS spectrophotometer (Madrid, Spain) equipped with Cintral Software, according to the literature references [40] with slight modifications. The content of the XTDs into TPP solution was evaluated spectrophotometrically at different wavelengths, corresponding to each derivative, as follows: λ = 276 nm for 6k, λ = 277 nm for 6c and 6e, λ = 280 nm for 6f. The recorded values were used to calculate the non-loaded and loaded amount of each XTD. A standard curve for each XTD with a specific correlation coefficient (y = 21.23x − 0.0112; R^2^ = 1 for 6c, y = 27.606x − 0.0151; R^2^ = 1 for 6e, y = 28.89x + 0.0219; R^2^ = 0.9996 for 6f, y = 20.914x + 0.0406; R^2^ = 1 for 6k) was used. 

The drug loading (DL) was calculated using the following formula [40]:(2)$$\mathrm{DL}\left(\%\right)=\frac{W_{3}}{W1}\times 100\%$$
where: W_1_ = the amount of CS-XTD microparticles, after drying (mg); W_3_ = the amount of XTD loaded into the chitosan matrix (mg).

The entrapment efficiency (EE) was calculated using the following formula [41]: (3)$$\mathrm{EE}\left(\%\right)=\frac{W_{3}}{W_{2}}\times 100\%$$
where: W_2_ = the mount of XTD used in study (mg); W_3_ = the amount of XTD loaded into chitosan matrix(mg).

All tests were performed in triplicate and the results are expressed as mean DL/EE ± standard deviation.

#### 2.3.4. In Vitro Release 

In vitro release of XTDs from polymeric system based chitosan (CS-XTDs: CS-6c, CS-6e, CS-6f, CS-6k) was performed using simulated gastric fluid (SGF, pH 1.2) and simulated intestinal fluid (SIF, pH 6.8), according to the literature references, with slight modifications [42,43,44]. The polymeric systems (CS-XTDs) were firstly placed in SGF for two hand then were moved to SIF. Briefly, a weighed amount of CS-XTDs microparticles was placed in a flask with 2 mL of media and incubated at 37 ± 0.1 °C under stirring (100 rpm). Every 30 min, a sample of 1.8 mL was collected from the media and replaced with an equal volume (1.8 mL) of fresh media. The concentration of the XTDs in the solution was evaluated spectrophotometrically at the specific wavelengths, corresponding to each derivative, using the standard curve previously obtained. 

The Drug release (DR) was calculated using the following formula [42]:(4)$$\mathrm{DR}\left(\%\right)=\frac{C_{1}}{C_{0}}\times 100\%$$
where: C_1_ = concentration of the XTD (mg/mL) in the release media at different times; C_0_ = concentration of the XTD (mg/mL) in the CS-XTD microparticles.

All tests were performed in triplicate and the results are expressed as mean DR ± standard deviation.

#### 2.3.5. Fourier Transform Infrared (FTIR) Spectroscopy 

FT-IR spectra of chitosan, XTDs (6c, 6e, 6f, 6k) and of CS-XTDs (CS-6c, CS-6e, CS-6f, CS-6k) were recorded using ABB-MB3000 FT-IR MIRacleTM Single Bounce ATR (Zürich, Switzerland) at a resolution of 4 cm^−1^, after 16 scans, in the 4000–650 cm^−1^ range and processed with the Horizon MBTM FT-IR Software (Horizon MB 3.1.29.5, LabCognition GmbH & Co. KG, Cologne, Germany). 

### 2.4. Biological Evaluation

#### 2.4.1. Acute Toxicity Assay 

The acute toxicity of the XTDs (6c, 6e, 6f, 6k) was evaluated by determining the lethal dose of 50 (LD_50_). The Swiss mice (22–35 g) were housed in polyethylene cages, at constant conditions: temperature of 24 ± 2 °C, humidity of 40–70%, and cycle of 12 h light and 12 h dark. The acclimatization of the animals to laboratory conditions was performed 7 days before the experiment, receiving standard food and water ad libitum. The animals were kept fasting for 24 h before starting the experiment. Each group (six mice/group) received XTDs, orally (p.o.), as suspensions in Tween 80 in doses ranging between 1500–6500 mg/kg b.w. The survival rate was determined at 1 day, 2 days, 3 days, 7 days, and 14 days after administration.

The lethal dose 50 (LD_50_) was determined by the Karber method using the following formula [45,46]:(5)$$LD_{50}=LD_{100}-\frac{\Sigma (a\times b)}{n}$$
where: *a* = the difference between two successive doses of the XTDs; *b* = the arithmetic average of the mice died from two successive series; *n* = the number of animals from each group; LD_100_ = the 100% lethal dose.

This study was performed according to the ethics guidelines on laboratory animal studies (Law no. 206/May 27, 2004, EU/2010/63-CE86/609/EEC) and with agreement (no 17826/2016) of the Ethics Committee for Animal Research of “Grigore T. Popa” University of Medicine and Pharmacy Iasi.

#### 2.4.2. Antibacterial/Antifungal Tests

The antibacterial and antifungal activity of XTDs and CS-XTDs^3^ (5 mg/mL) were evaluated using the agar disc diffusion and broth micro-dilution methods.

##### The Agar Disc Diffusion Method.

Antibacterial and antifungal activity of XTDs and CS-XTDs, expressed as the diameter of inhibition area, were evaluated using the standard disk diffusion assay according to literature reference [47] with slight modifications. Prior to the test, the bacterial and yeast strains were diluted in NaCl (0.9 %) in order to achieve a turbidity equivalent of 10^6^ CFU/mL (McFarland standard no 5). The suspensions were diluted 1:10 in Mueller Hinton agar (bacteria) and Sabouraud agar (yeasts) and then spread on sterile Petri plates (25 mL/Petri plate). Sterile stainless steel cylinders (10 mm height and 5 mm internal diameter) were placed on the agar surface in Petri dishes. In each cylinder 200 µL of XTDs, solutions in DMSO (20 mg/mL) and an equivalent concentration of CS-XTDs were added. Commercial discs containing ampicillin (25 µg/disc), ciprofloxacin (30 µg/disc), and nystatin (100 µg/disc) were used as a positive control. DMSO was used as a negative control. The plates were incubated for 24 h at 37 °C (bacteria) and for 48 h at 24 °C (yeasts). After incubation the diameters of inhibition were measured. All tests were performed in triplicate and the results are expressed as mean diameter ± standard deviation.

##### The Broth Micro-Dilution Method.

The minimum inhibitory concentration (MIC) and the minimum bactericidal/fungicidal concentration (MBC/MFC) of XTD were determined using the standard two-fold dilution method standardized by [47], with slight modifications. The strains were inoculated in the specific agar medium and incubated at 37 °C for 24 h for bacteria, respectively 48 h at 24 °C for *Candida* strains. After the incubation, the culture media were diluted in order to obtain a final concentration of 10^6^ CFU/mL. In a 96-well microplate, different dilutions were prepared from the stock solution of the XTDs (20 mg/mL), in order to obtain a volume of 100 μL in each well with final concentrations in the range of 10 mg/mL, 5 mg/mL, 2.5 mg/mL, 1.25 mg/mL, 0.625 mg/mL, 0.312 mg/mL, 0.156 mg/mL, 0.078 mg/mL, 0.039 mg/mL, 0.0195 mg/mL, 0.009 mg/mL, and 0.0048 mg/mL. A volume of 100 μL strain suspension was inoculated onto the wells and the microplates were incubated for 24 h at 37 °C (bacterial strains) and at 24 °C for (fungal strains). The MIC value was established as the lowest concentration, which determined the visual inhibition of the strain’s growth. For MBC/MFC determination, 1 µl from each visually complete inhibition was transferred onto a plate with solid media and incubated for 24 h at 37 °C (bacterial strains) and at 24 °C for (fungal strains). The MBC/MFC values were considered the lowest concentration, which killed 99.9% of the tested strains. All tests were performed in triplicate.

## 3. Results and Discussions

The synthesis of new xanthine derivatives with thiazolidine-4-one scaffold (6a–k) (Figure 1) were presented in our previous paper [38]. Some of these compounds showed good antioxidant effects in terms of the antiradical scavenging effect (DPPH, ABTS) and phosphomolybdenum reducing antioxidant power [38]. In this study the compounds were evaluated for antibacterial and antifungal effects and the most active of them were formulated as new polymeric systems based on chitosan in order to increase their biological effects. 

### 3.1. New Polymeric Systems Based on Chitosan

The chitosan microparticles could be prepared using chemical and physical methods. The chemical method is based on the covalent bond formation between the functional groups of chitosan while the physical process involves electrostatic hydrophobic interactions or hydrogen bonds formation in the polymer matrix [48,49]. It is considered that the last type of polymeric system seems to have increased biocompatibility and are more tolerated than the chemical ones [50,51]. The most used physical method is ionic gelation, which is based on the ionic complexes formation between positively charged amino groups and anions such as sulphate, citrate, and phosphate. The most used cross-linking agent is pentasodiumtripolyphosphate, which interacts with the amino groups of chitosan by electrostatic forces [52]. 

Four XTDs (6c, 6e, 6f, 6k) were selected based on their biological effects and were inglobated into a polymeric matrix based on chitosan using the ionic gelation method. In order to optimize the preparation procedure of chitosan microparticles loaded with xanthine derivatives with thiazolidine-4-one scaffold (CS-XTDs) different parameters were applied: chitosan concentration (1.2%–1.7%), TPP concentration (2%–3%), and reticulation time (4–8 h). The most stable CS-XTDs systems were obtained using the following parameters: CS concentration of 1.7%, TPP concentration of 2%, and reticulation for 8 h. Using three concentrations (3 mg/mL, 4 mg/mL, 5 mg/mL) for each XTD, twelve CS-XTD systems were developed and characterized. The mass ration between CS, TPP, and XTD, based on the concentration of used XTD (9 mg, 12 mg, 15 mg) was as follows:5.7:44:1, 4.25:33:1 and 3.4:27:1.

#### 3.1.1. Particle Size and Morphology

The size of the CS-XTDs ranged between 858–953 μm (wet state) and 611–855 μm (dry state) and it was higher than CS (825 μm—wet state; 611 μm—dry state), thus confirming the loading process (Table 1). The data showed that the size increases with the concentration of the compounds and are in good agreement with similar data presented in the literature [53].

The SEM showed that chitosan microparticles (CS) have a spherical and regular shape, regular outline and a slightly rough surface. Upon loading of XTD, a deformation of microparticles was observed, which is intensified by the increasing concentration described in other studies [5]. The most intense deformation was observed in the case of CS-6c^3^ (5 mg/mL), for which the spherical shape is lost and pronounced flattening was recorded (Figure 2).

The results showed that a higher concentration of the cross-linking agent was associated with decreased stability of microparticles during the gelation process. Also, it was highlighted that the deformation of CS-XTDs microparticles during the air-drying process was directly proportional to the concentration of the TPP solution, which resulted in strongly flattened microparticles being obtained for a value of 3% TPP. The reticulation time is also an important parameter for the stability of microparticles; increasing the time was associated with the increased stability of the developed polymeric systems [54,55].

#### 3.1.2. Swelling Degree (SD)

The swelling degree is an important parameter in the release of the drug from the polymer matrix, having an important influence on the drug’s bioavailability. The results for CS-XTDs systems recorded in distilled water and simulated gastric fluid (SGF, pH 2.2) are presented in Figure 3, Figure 4, Figure 5 and Figure 6. For distilled water, the swelling degree values of CS-XTDs was comparable with that of CS. The CS-XTDs absorbed the highest amount of water in the first 10 mins of the experiment, with the SD (%) value ranging around 150% (Figure 3 and Figure 4). This phenomenon could be explained by the cross-linking agent which, by dissolution in aqueous medium, released the amino groups of chitosan, increasing the hydrophilic character of the polymer matrix. 

The thermodynamic equilibrium state was reached after different times depending on the structure of XTDs loaded into the polymer matrix. At the end of the experiment the swelling degree values ranged between 190 ± 2.6 (CS-6e^3^) and 206 ± 2.9 (CS-6k^1^). In similar conditions the recorded value for CS was 209 ± 3.5 (Table 2).

The analysis of the SD results recorded in SGF highlighted that the SD of CS-XTDs was lower than of CS (375%) (Figure 5 and Figure 6); the results could be explained based on the hydrophobic character of the XTD loaded into the polymer matrix, a character which results in decreasing the chitosan matrix permeability and thus the swelling degree. The presence of functional groups such as halogen (6c) or methyl (6k), attached to the aromatic ring of the thiazolidine-4-one scaffold increases the hydrophobic character of the XTD. Our observation is in agreement with other literature data [56]. A high absorption degree (over 300%) at the beginning of the experiment was observed, after which it decreased in intensity. For CS the thermodynamic equilibrium state was reached after 2 h and maintained until the end of the experiment. In similar conditions, for CS-XTDs the thermodynamic equilibrium state was reached after 40 min–2 h, depending on the kind of XTD loaded into the polymer matrix, which could result in a shorter period. 

#### 3.1.3. Drug Loading and Entrapment Efficiency

The results obtained for drug loading (DL) and entrapment efficiency (EE) parameters are presented in Figure 7 (CS-6c, CS-6e) and Figure 8 (CS-6f, CD-6k). The study highlighted that the entrapment efficiency ranged between 63.71% (CS-6e^1^) and 93.91% (CS-6k^3^) and is directly proportional to the concentration. The highest entrapment efficiency, at all three concentrations (3 mg/mL, 4 mg/mL, 5 mg/mL), was recorded for CS-6k, the values being: 87.86 ± 1.25% (3 mg/mL), 91.58 ± 0.55% (4 mg/mL) and 93.91 ± 1.41% (5 mg/mL). These results could be explained based on the structure of the XTD (6k), which has hydrophobic *methyl* group at *para* position of the aromatic ring. The XTD with hydrophilic *methoxy* group attached at the *ortho* (6e) and *para* (6f) position on the aromatic ring have showed lower values of entrapment efficiency.

Concerning the DL of XTDs into the polymeric matrix, it was noted that this parameter increases with the concentration. The obtained values ranged between 10.04 ± 0.49% (CS-6e^1^) and 22.02 ± 0.74% (CS-6k^3^). As a previous parameter, the highest values were obtained for derivative 6k, which has a *methyl* group at the *para* position of the aromatic ring as follows: 14.32 ± 1.12% (CS-6k^1^), 18.90 ± 0.67% (CS-6k^2^), and 22.02 ± 0.74% (CS-6k^3^). A good correlation was also observed between DL and EE values, which proves the accuracy of the study. Thus, it has been demonstrated that the optimal CS-XTD formulation was obtained for 5 mg/mL of XTD.

#### 3.1.4. In Vitro Drug Release 

The literature describes two methods for releasing the drugs from the polymer matrix: by direct and indirect diffusion, the last one being based on the dissolution of the drug into the matrix, followed by the release through membrane pores. In turn, the membrane permeability is influenced by the method used to prepare the polymeric systems, by the morphology of microparticles, and not least by the chemistry structure of the drug. The oral route is the most used for drug administration. After the absorption of drugs through gastric or intestinal mucosa, it will pass to the blood system and then will arrive at the site of action. Normally, the gastric pH ranges between 1 and 1.5 and will contain 99% water and only 1% different organic or inorganic substances, while the small intestine has a pH, which ranges between 4.8 to 8.2.

In the SGF, the drug release from CS-XTD was ranging between 51.34% and 98.42%, while the drug release in SFI was lower, being around 5%, excepting the CS-6k for which the drug release in SIF was 17% (Figure 9 and Figure 10). The lowest cumulative release was observed for CS-6f (CS-6f^1^ = 58.10%, CS-6f^2^ = 51.34%, CS-6f^3^ = 54.52%), while the highest values were recorded for CS-6c (CS-6c^1^ = 98.42%, CS-6c^2^ =89.92%, CS-6c^3^ = 97.44%). 

#### 3.1.5. FTIR Analysis

The confirmation of the loading process was performed by highlighting in FTIR spectra the CS-XTDs of the specific functional groups of the components of the polymeric matrix: CS, XTDs, and TPP. The FTIR spectra of CS-XTDs (CS-6c, CS-6e, CS-6f and CS-6k) at different concentrations in reference with the FTIR spectra of CS and XTDs (6c, 6e, 6f and 6k) are presented in Figure 11.

In acetic acid solution, chitosan has a pKa of 6.3 and a polycationic structure with positively charged amino groups. In water, TPP dissociates to form both hydroxyl and phosphoric ions. In the crosslinking process the cationic amino groups of chitosan react with negatively charged TPP to form ionic complexes by electrostatic bonds. A secondary mechanism based on hydrogen-hydrogen bonds between the hydroxyl groups of CS and of TPP is also possible [50]. The XTDs are physically and uniformly dispersed into the polymer matrix. The concentration of CS and XTDs are very important parameters, because they influence the physic-chemical properties of the CS-XTD systems [49].

The specific vibrations of functional groups of CS have been identified in the following regions: 3362–3285 cm^−1^ (OH, NH), 2981–2872 cm^−1^, 1431–1414 cm^−1^ and 1377–1375 cm^−1^ (–CH_2_–CH_3_), 1661–1647 cm^−1^ (C=O), 1267–1250 cm^−1^ (C–N), 1034–1026 cm^−1^ (C–O–C). The presence of XTDs (6c, 6e, 6f and 6k) into the polymer matrix was confirmed by identification of the specific absorption bands: NH bond (3362–3285 cm^−1^), thiazolidine-4-one ring: 2981–2872 cm^−1^ (–CH_2_–), 1724–1711 cm^−1^ (C=O) and 671–660 cm^−1^ (C–S), aromatic ring: 3362–3285 cm^−1^, 810–746 cm^−1^ (=C–H), 1558–1549 cm^−1^ and 1494–1452 cm^−1^ (C=C), amide group, which are overlapping with valence vibrations of the same group in the chitosan structure, and not the substituents on the aromatic ring at 1063–1059 cm^−1^ (C–O from –OCH_3_ in case of 6e and 6f), 822–818 cm^−1^ (–Cl in case of 6c) and 1377–1375 cm^−1^ (–CH_3_ in case of 6e, 6f and 6k). The reticulation agent, TPP, was identified by specific absorption bands in the 1151–1149 cm^−1^ region (P=O) and the 895–893 cm^−1^ region (P–O–P).

### 3.2. Biological Evaluation

#### 3.2.1. Toxicity Degree

The toxicological screening highlighted that all tested XTDs (6c, 6e, 6f, 6k) have a low toxicity, with the following LD_50_ values: 2125 mg/kg bw (6c), 1687.5 mg/kg bw (6e), 1937.5 mg/kg bw (6f) and 1312.5 mg/kg bw (6k). These compounds proved to be less toxic than theophylline, which is used as a starting reagent in their synthesis (LD_50_ = 332 mg/kg bw) [57] and supports the favorable influence of thiazolidine-4-one scaffold upon the chemical modulation of xanthine structure.

#### 3.2.2. Antibacterial/Antifungal Study

The data presented in Table 3 showed that XTDs (6a–k) and CS-XTDs^3^ (CS-6c^3^, CS-6e^3^, CS-6f^3^, CS-6k^3^) are active on both bacterial and fungal strains, the diameter of the inhibition area being in correlation with structure of the XTD and also with bacterial and yeasts strains, respectively. 

Referring to the XTDs it was noted the most active compound on *Staphylococcus aureus* ATCC 25923 was 6d, for which the diameter of the inhibition area was 17.1 ± 0.24 mm. The compounds 6a, 6f and 6g showed improved antibacterial effects on *Sarcina lutea* ATCC 9341, their effect (6a: 20.1 ± 0.17 mm, 6f: 20.1 ± 0.43 mm, 6g: 20.0 ± 0.31 mm) being higher than the effect of CS and comparable with ciprofloxacin (25.0 ± 0.1 mm). The effect of XTDs on Gram negative bacterial strain *Escherichia coli* ATCC 25922 was reduced in reference to Gram positive bacterial strains (*Staphylococcus aureus, Sarcina lutea*), the diameter of the inhibition area ranging between 10.4 mm and 12.2 mm.

Appreciable antifungal effects were noted for compounds 6c and 6h on *Candida albicans* ATCC 10231 (15.2 ± 0.21 mm and 15.2 ± 0.26 mm respectively), for 6h and 6g on *Candida glabrata* ATCC MYA 2950 (19.1 ± 0.17 mm and 16.1 ± 0.28 mm respectively) and for 6f and 6g on *Candida parapsilosis* ATCC 22019 (15.2 ± 0.11 mm and 15.4 ± 0.31 mm respectively). 

As we expected, the loading of the XTDs into the polymer matrix based on chitosan was associated with increasing the antimicrobial effects, all CS-XTD systems (CS-6c, CS-e, CS-6f, CS-6k) show improved antibacterial and antifungal effects in reference to corresponding XTD (6c, 6e, 6f, 6k) (Table 3).

The antibacterial effect of 6d was confirmed by the broth micro-dilution assay, for which the minimum inhibitory concentration (MIC) and minimum bactericidal concentration (MBC) were 0.3125 mg/mL and 10 mg/mL, respectively (Table 4). The similar results were obtained also for 6a. It was also noted that, in most cases, the value of MBC was 10 mg/mL, higher than the value of MIC, which means that the compounds act, especially, by the inhibition of microbial growth that can then act as bactericidal agents.

## 4. Conclusions

In this study new polymeric systems based on chitosan (CS) loaded with new xanthine derivatives with thiazolidine-4-one scaffold (XTDs) were developed and characterized. The success of the loading process was proved by highlighting in FTIR spectra of the CS-XTDs of the specific functional groups of XTD. The optimized polymeric systems were evaluated in terms of particle size, morphology, swelling degree, drug loading and entrapment efficiency. The results demonstrated a good swelling degree for CS-XTDs and the entrapment efficiency of the XTD into polymer matrix was between 63% and 94%. In the simulated biological fluids (SGF, SIF), an increased cumulative release, between 54.52% and 97.44% was observed. The data supports also improved antimicrobial effects for CS-XTDs in reference with XTDs. Based on the obtained results the developed polymeric systems could have important applications in the food field as active multifunctional (antimicrobial, antifungal) packaging materials and in medical and pharmaceutical fields. 

## Figures and Tables

**Figure 1 materials-12-00558-f001:**
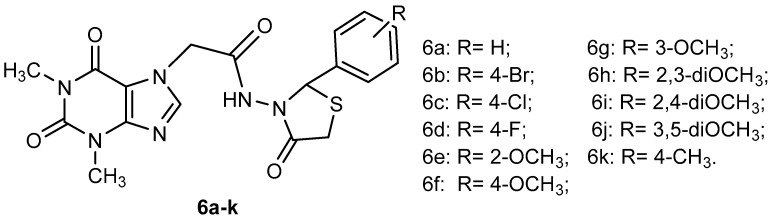
The structure of the xanthine derivatives with thiazolidine-4-one scaffold (XTDs: 6a-k).

**Figure 2 materials-12-00558-f002:**
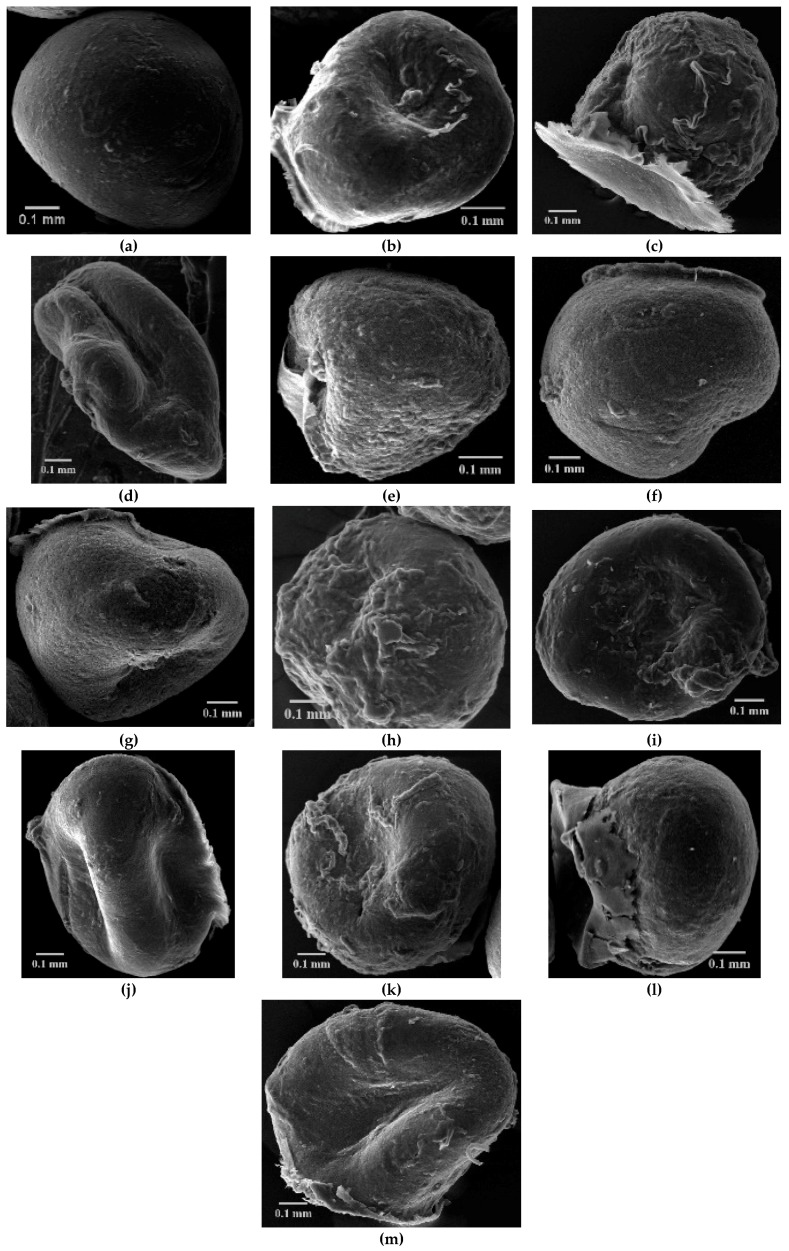
Scanning electron microscope (SEM) images for CS and CS-XTDs microparticles (**a**: CS, **b**: CS-6c^1^, **c**: CS-6c^2^, **d**: CS-6c^3^, **e**: CS-6e^1^, **f**: CS-6e^2^, **g**: CS-6e^3^, **h**: CS-6f^1^, **i**: CS-6f^2^, **j**: CS-6f^3^, **k**: CS-6k^1^, **l**: CS-6k^2^, **m**: CS-6k^3^); ^1^ = 3 mg/mL, ^2^ = 4 mg/mL, ^3^ = 5 mg/mL.

**Figure 3 materials-12-00558-f003:**
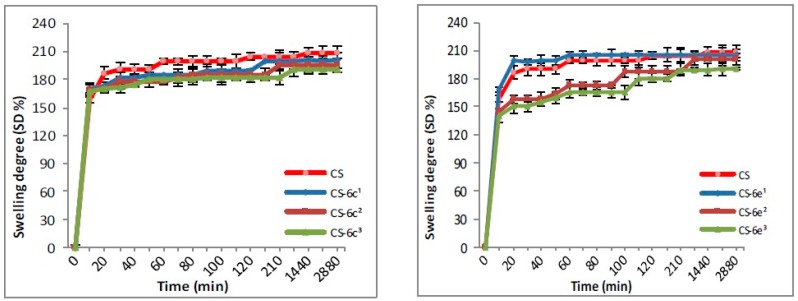
Swelling degree of CS and CS-XTDs (CS-6c; CS-6e) in distilled water at different concentrations (^1^ = 3 mg/mL, ^2^ = 4 mg/mL, ^3^ = 5 mg/mL).

**Figure 4 materials-12-00558-f004:**
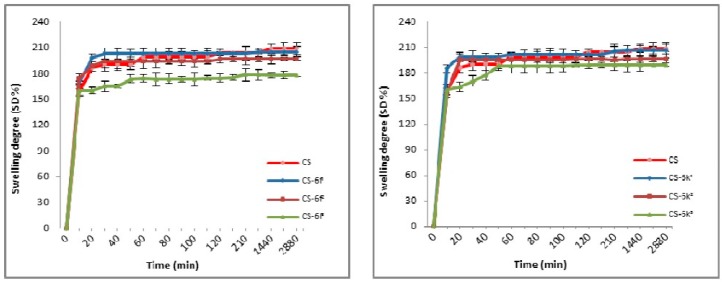
Swelling degree of CS and CS-XTDs (CS-6f; CS-6k) in distilled water at different concentrations (^1^ = 3 mg/mL, ^2^ = 4 mg/mL, ^3^ = 5 mg/mL).

**Figure 5 materials-12-00558-f005:**
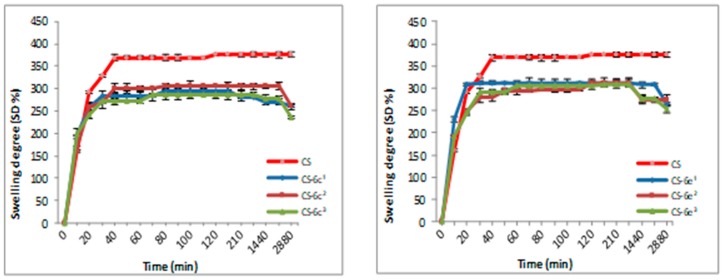
Swelling degree of CS and CS-XTDs (CS-6c; CS-6e) in SGF at different concentrations (^1^ = 3 mg/mL, ^2^ = 4 mg/mL, ^3^ = 5 mg/mL).

**Figure 6 materials-12-00558-f006:**
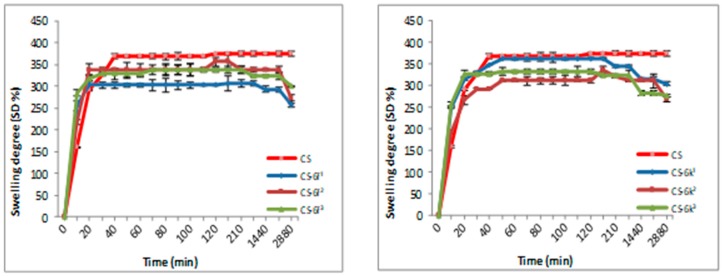
Swelling degree of CS and CS-XTDs (CS-6f; CS-6k) in SGF at different concentrations (^1^ = 3 mg/mL, ^2^ = 4 mg/mL, ^3^ = 5 mg/mL).

**Figure 7 materials-12-00558-f007:**
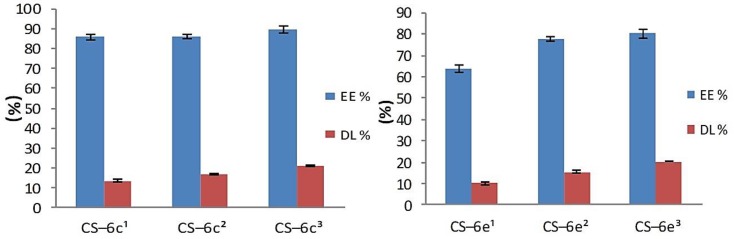
The Entrapment efficiency (EE%) and Drug loading (DL%) for CS–6c andCS-6e (^1^ = 3 mg/mL, ^2^ = 4 mg/mL, ^3^ = 5 mg/mL).

**Figure 8 materials-12-00558-f008:**
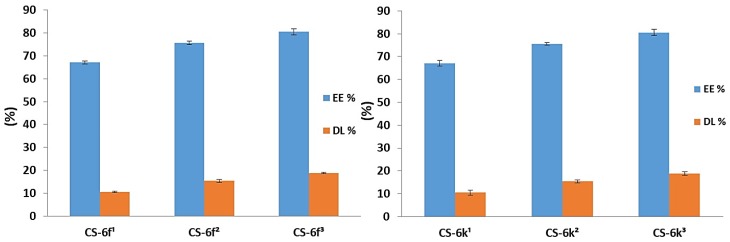
The Entrapment efficiency (EE%) and Drug loading (DL%) for CS–6f and CS-6k (^1^= 3 mg/mL, ^2^ = 4 mg/mL, ^3^ = 5 mg/mL).

**Figure 9 materials-12-00558-f009:**
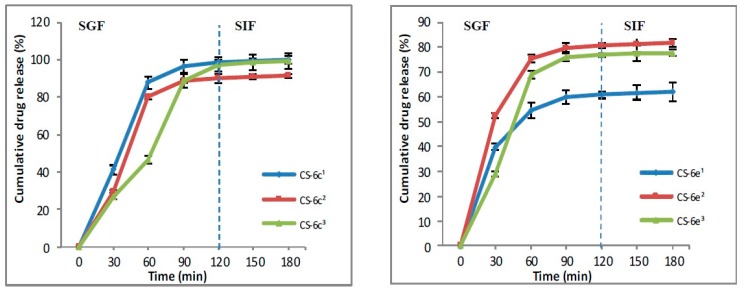
The release profile of compound 6c and 6e from CS-6c and CS-6e respectively (^1^ = 3 mg/mL, ^2^ = 4 mg/mL, ^3^ = 5 mg/mL).

**Figure 10 materials-12-00558-f010:**
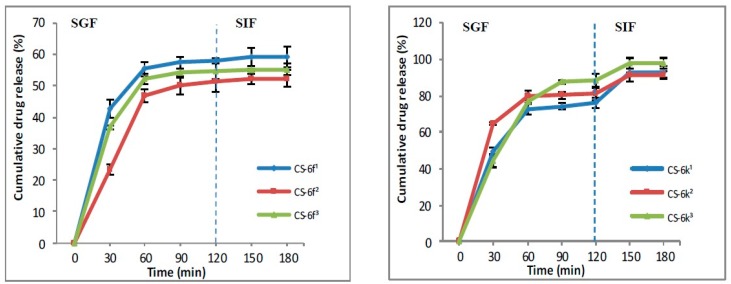
The release profile of compound 6f and 6k from CS-6f and CS-6k respectively (^1^ = 3 mg/mL, ^2^ = 4 mg/mL, ^3^ = 5 mg/mL).

**Figure 11 materials-12-00558-f011:**
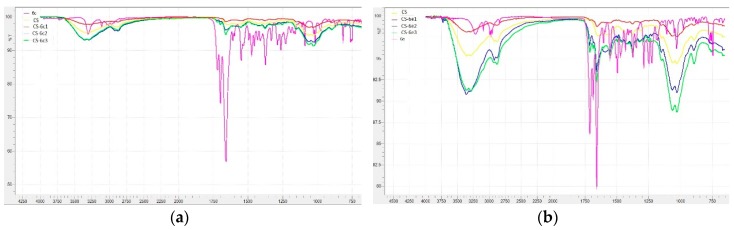
The Fourier transform infrared (FTIR) spectra of CS, XTDs (6c, 6e, 6f, 6k) and CS-XTDs (CS-6c, CS-6e, CS-6f, CS-6k) (^1^ = 3 mg/mL, ^2^ = 4 mg/mL, ^3^ = 5 mg/mL).

**Table 1 materials-12-00558-t001:** Microparticles size of CS and CS-XTDs (CS-6c, CS-6e, CS-6f, CS-6k) at different concentrations.

CS/CS-XTDs	Size in Wet State (µm)	Size in Dry State (µm)	CS-XTDs	Size in Wet State (µm)	Size in Dry State (µm)
CS-6c^1^	904 ± 4.2	638 ± 3.1	CS-6f^1^	887 ± 2.1	614 ± 3.1
CS-6c^2^	897 ± 7.2	762 ± 2.5	CS-6f^2^	921 ± 4.0	675 ± 5.0
CS-6c^3^	951 ± 3.1	855 ± 2.1	CS-6f^3^	947 ± 3.2	751 ± 6.4
CS-6e^1^	858 ± 3.1	632 ± 3.5	CS-6k^1^	929 ± 6.0	639 ± 7.0
CS-6e^2^	923 ± 3.5	634 ± 4.7	CS-6k^2^	912 ± 5.0	688 ± 4.0
CS-6e^3^	953 ± 2.9	646 ± 4.0	CS-6k^3^	918 ± 2.1	728 ± 4.7
CS	825 ± 3.5	611 ± 3.1	-	-	-

^1^ = 3 mg/mL, ^2^ = 4 mg/mL, ^3^ = 5 mg/mL.

**Table 2 materials-12-00558-t002:** Swelling degree of CS and CS-XTD at thermodynamic equilibrium state.

CS-XTDs	SD (%)	CS-XTDs	SD (%)	CS-XTDs	SD (%)	CS-XTDs	SD (%)
CS-6c^1^	200 ± 2.7 *	CS-6e^1^	205 ± 3.6	CS-6f^1^	205 ± 4.5	CS-6k^1^	206 ± 2.9
CS-6c^2^	195 ± 2.2 *	CS-6e^2^	201 ± 1.7 *	CS-6f^2^	197 ± 3.6 *	CS-6k^2^	196 ± 5.2 *
CS-6c^3^	190 ± 4.1 *	CS-6e^3^	190 ± 2.6 *	CS-6f^3^	178 ± 3.1 *	CS-6k^3^	189 ± 3.1 *
CS	209 ± 3.5	-	-	-	-	-	-

* significant different (*p* < 0.05) in reference with CS; ^1^ = 3 mg/mL, ^2^ = 4 mg/mL, ^3^ = 5 mg/mL.

**Table 3 materials-12-00558-t003:** Antibacterial/antifungal inhibition area (mm) of XTDs (6a–k) and CS-XTDs.

Sample	Diameter of Inhibition Area*(mm)					
	Bacterial Strains			Yeasts Strains		
	SA	SL	EC	CA	CG	CP
6a	15.2 ± 0.23	20.1 ± 0.17	12.2 ± 0.35	11.1 ± 0.36	12.2 ± 0.22	12.2 ± 0.24
6b	15.3 ± 0.40	19.1 ± 0.28	11.1 ± 0.23	10.1 ± 0.21	10.8 ± 0.37	10.3 ± 0.31
6c	14.9 ± 0.21	19.0 ± 0.13	10.4 ± 0.15	15.2 ± 0.21	12.1 ± 0.42	11.2 ± 0.24
6d	17.1 ± 0.24	18.1 ± 0.42	11.2 ± 0.47	10.0 ± 0.11	11.0 ± 0.46	11.3 ± 0.32
6e	14.0 ± 0.38	18.8 ± 0.29	11.1 ± 0.14	10.1 ± 0.28	10.9 ± 0.26	14.2 ± 0.12
6f	15.0 ± 0.50	20.1 ± 0.43	11.2 ± 0.35	12.1 ± 0.07	10.0 ± 0.06	15.2 ± 0.11
6g	15.5 ± 0.27	20.0 ± 0.31	12.1 ± 0.26	11.3 ± 0.23	16.1 ± 0.28	15.4 ± 0.31
6h	15.4 ± 0.34	19.2 ± 0.09	12.0 ± 0.06	15.2 ± 0.26	19.1 ± 0.17	12.0 ± 0.14
6i	15.4 ± 0.25	19.1 ± 0.11	12.1 ± 0.04	12.5 ± 0.10	15.0 ± 0.24	12.1 ± 0.23
6j	15.2 ± 0.14	19.1 ± 0.17	12.2± 0.26	12.2 ± 0.26	15.3 ± 0.09	12.1 ± 0.24
6k	14.3 ± 0.19	17.8 ± 0.23	11.0 ± 0.31	10.3 ± 0.23	13.2 ± 0.11	11.0 ± 0.13
CS-6c^3^	16.4 ± 0.41	21.0 ± 0.23	12.3 ± 0.21	17.4 ± 0.29	14.3 ± 0.38	13.2 ± 0.31
CS-6e^3^	15.2 ± 0.23	20.4 ± 0.21	13.4 ± 0.25	12.5 ± 0.21	12.6 ± 0.21	15.4 ± 0.26
CS-6f^3^	21.2 ± 0.43	25.1 ± 0.28	14.7 ± 0.38	16.7 ± 0.42	12.3 ± 0.51	18.4 ± 0.42
CS-6k^3^	17.1 ± 0.32	22.4 ± 0.18	15.2 ± 0.18	14.6 ± 0.21	16.3 ± 0.38	17.2 ± 0.35
CS	12 ± 0.35	11 ± 0.26	9 ± 0.41	-	-	-
C^a^	25.1 ± 0.08	25.0 ± 0.1	28.9 ± 0.18	-	-	-
A^b^	27.1 ± 0.12	31.8 ± 0.15	21.0 ± 0.21	-	-	-
N^c^	-	-	-	20.1 ± 0.11	21.0 ± 0.14	20.1 ± 0.09

* mean values (n = 3) ± standard deviation. SA = *Staphylococcus aureus* ATCC 25923; SL = *Sarcina lutea* ATCC 9341; EC = *Escherichia coli* ATCC 25922; CA = *Candida albicans* ATCC 10231; CG = *Candida glabrata* ATCC MYA 2950; ^3^ = 5 mg/mL; CP = *Candida parapsilosis* ATCC 22019; C^a^= Ciprofloxacin (30 µg/disc); A^b^ = Ampicillin (25 µg/disc); N^c^ = Nystatin (100 µg/disc).

**Table 4 materials-12-00558-t004:** Antimicrobial activity expressed as MIC and MBC values (mg/mL) of XTDs (6a–k).

Sample	*Staphylococcus Aureus* ATCC 25923		*Escherichia Coli* ATCC 25922		*Candida Albicans* ATCC 90028	
	MIC *	MBC *	MIC *	MBC *	MIC *	MBC *
6a	0.625	10	1.25	1.25	1.25	1.25
6b	2.5	10	2.5	2.5	1.25	1.25
6c	2.5	10	1.25	1.25	1.25	2.5
6d	0.3125	10	1.25	2.5	2.5	2.5
6e	1.25	10	1.25	1.25	1.25	2.5
6f	0.625	10	1.25	2.5	2.5	2.5
6g	0.625	10	1.25	2.5	2.5	2.5
6h	1.25	5	2.5	2.5	2.5	2.5
6i	0.625	10	2.5	2.5	2.5	2.5
6j	0.625	10	1.25	2.5	2.5	2.5
6k	1.25	10	0.625	2.5	0.625	1.25
A	1 ^1^	2 ^1^	2 ^1^	4 ^1^	-	-
N	-	-	-	-	8 ^1^	16 ^1^

* mean values (n = 3) ± standard deviation; ^1^ µg/mL.

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
