# Peer review of "Formulation and Characterization of New Polymeric Systems Based on Chitosan and Xanthine Derivatives with Thiazolidin-4-One Scaffold"

_materials, 2019, doi:10.3390/ma12040558_

Reviewer 1 Report

This paper, titled “Formulation And Characterization Of New Polymeric Systems Based Chitosan And Xanthine Derivatives With Thiazolidin-4-One Scaffold” review comments are as follows:

In the abstract, the results of the experiment should be explained in detail and should be revised. For example, antidiabetic properties, antioxidant did not see relevant experiments. The particle size coverage and the most effective results should be indicated and indicated by scientific notation.

What is the molecular weight of the chitosan material?

In Page3, line 120 “different amounts of XTD were dissolved in appropriate volume of DMSO” The dose used in the experiment should be detailed.

Line 114, 124“under light shaking, using a syringe needle (26 G; 0.45 mm x 16 mm)” The experimental method description is not clear

Line 220,235, 106 CFU/mL or 1E+6 CFU/mL?

Line 265-271, Describe the results, we still do not understand, the importance of the experiment and its detailed practice, or the ratio between chitosan and TPP, what are the advantages and disadvantages of the amount formed?

Line 272-278, What is the ratio between CS and TPP and XTD?

In Table 2, What is the purpose of the experiment with different doses (3 mg/mL, 4 mg/mL, 5 mg/g)? What is the difference between the experimental results?

Figure 2 should be placed in a scale bar.

In Figure 3-4, the swelling degree decreases as the dose of XTD is increased. What is the meaning of using different XTD doses.

The chitosan content is affected by the pH value. What is the swelling degree under neutral to basic pH conditions?

In Figure 3-4 and Table 2, the upper and lower marks of the numbers should be consistent.

In Table 2, is there a statistically significant difference?

Line323, than of CH (375%) (Figure 5, 6)? Or CS?

Line322-328 What are the characteristics of different compounds, such as functional groups, and what role does it play in the swelling degree?

Line336-351, what are the EE and DL rankings of different XTD compounds, and what are the differences in the functional groups? What kind of molecular bond strengths are involved?

Table 3, The antibacterial activity of chitosan alone can be added to the table.

Author Response

Dear Sir,

Thank you for your time and recommendations. All the requests have been made in the manuscript (in red color). For  answers to your comments/questions/suggestions please see the attached document.

Reviewer 2 Report

Remedial plants are the basis for the disclosure of various commercial natural drugs including a xanthine (XT) family with the typical representatives namely theophylline, theobromine, caffeine, etc. To enhance the biological activity of the common drugs, the authors synthesized the original group of remedies as chemical products of   XT and thiazolidine-4-one scaffold (XTD) that are embedded into chitosan microparticles. The following complex investigations such as microscopic geometry registration, swelling measurement, FTIT, SEM, and bio-functional study via antimicrobial/antifungal assay and toxicity evaluation have preliminarily shown the efficacy of novel drug therapeutic systems and possibly active packaging systems. In the next ensuing publications, the authors should explore barrier characteristics or other special functions of given materials as well.

The paper introduction provides a coherent laconic background of the topic that immediately gives to the reader perception awareness on the necessity conjugate design. The motivation and the objectives for performing the study are clearly defined in this section. All parts of the paper presented in a fairly concise manner and successively related among themselves. 

The title of the manuscript should be amended: in the phrase “base on” the preposition “on” is omitted  

I strongly recommend introducing the list of abbreviations dispersed through all text

I would like to suggest the authors insert additional arguments concerning with the application of novel materials for packaging. Please supply the readers with information on package perspectives e.g. in the food industry.  

L 375 (Section 3.1.5)  The existence of three components (chitosan, modified xanthine, and TPP) has been verified by FTIR technique, however their interactions as the evidence of mechanism(s) immobilization /encapsulation was not shown. It is worth to specify and partly expand this section to enhance the discussion of drug entrapping mechanism or present the additional information from the literature.

Fig 2.  I am not sure that the globules in the pictures look like transparent. Please pay attention again and explicate.

259, 260: “it was considered that the last type of polymeric systems seems have increased biocompatibility and are more tolerated than the chemical ones” [Physical encapsulation is more preferable relative to a chemical one – I agree]. Please supply this important statement with some reference(s). 

Additionally, I detected several grammar typos

 347: this parameter increaseS with ..

 357 and 399: the disSolution of the drug in(to) the matrix  

 Please check the text carefully again.

  The content of this paper with appropriate terminology and adequate argumentation falls within the scope of the materials and after making the above corrections, I recommend the manuscript for further performing in the Journal as an essential impact upon the Special Issue.

Author Response

(The authors gave the same response as above.)

Reviewer 3 Report

Authors should change the title as well as where they refer new, since neither the synthesis nor the method of encapsulation is new.
The results were only presented and need to be discussed in comparison with literature data.

As it is written on line 101 this not a new scaffold! The title of the article needs to be reformulated in order do not stat is new!

Line 101 …synthesized by our research group and were reported in previous paper…

Correct the names of bacteria and fungi for italic type characters!

Line 102 (Staphyloccoccus aureus ATCC 25923, Sarcina lutea ATCC 9341, Escherichia coli ATCC 25922) Line 103and  yeast strains (Candida albicans ATCC 10231, Candida glabrata ATCC MYA 2950, Candida parapsilosis ATCC 22019)

Please correct to 106 CFU/mL and correct throughout the text

Line 220 …% NaCl until the turbidity was equivalent to McFarland standard no. 0.5 (106 CFU/mL)…

The plural of medium is media! Please correct in all text!

Line 234….Candida strain. After the incubation, the mediums ….

According the norm that you describe to follow for the determination of the minimum inhibitory concentration (MIC) the cell density must be 5 × 105 CFU/mL (or 5 × 104 CFU/well in the microdilution method).and you used much higher cell density 106 CFU/mL!

Line 219 Clinical & Laboratory Standards Institute (CLSI) [44]

The described production of chitosan microparicles are not new! Please revise the title in 3.1

Line 255 3.1. New polymeric systems based on chitosan

In line 276 you refer stability! How did you evaluate that?

Line 276 ….The most stable CS-XTD systems were obtained in the similar conditions of CS microparticles.

Which was the procedure of particle drying?

Line 297…. deformation of CS-XTD microparticles during the drying process

Please added data or bibliographic support of your findings that the reticulation time is important on the stability!

Line 299 ….The reticulation time is also an important parameter for stability of microparticles, the increasing of the time being associated with increased stability…

Please replace in line 360 to oral way to oral route

Line 360…. The oral way is the most used for administration of drugs.

Author Response

Dear Sir,

Thank you for your time and recommendations. All the requests have been made in the manuscript (in red color). For  answers to your comments/questions/suggestions please see the attached document.

Round  2

Reviewer 3 Report

The authors answered all the questions and thus improved the article.

Author Response

Dear Reviewer,

Thank you for reviewer of the manuscript, for your time and expertise.

Best regards,

Lenuta Profire